# Failure Analysis of Chromium Plating Layer on the Surface of the Piston Rod of the Hydraulic Jack

**Qiankun Zhang [1], Jiangang Wang [1,2,*], Wenjing Shen [1], Fengshan Huang [3,*] and Yongjie Zhao [4]**

[1]  College of Material Science and Engineering, Hebei University of Science and Technology, Shijiazhuang 050018, China; zhangqiankun827@163.com (Q.Z.); shenwenjing0206@163.com (W.S.)

[2]  Key Laboratory of Near Net Forming of Materials in Hebei Province, Hebei University of Science and Technology, Shijiazhuang 050018, China

[3]  College of Mechanical Engineering, Hebei University of Science and Technology, Shijiazhuang 050018, China

[4]  Hebei Qianjin Machinery Factory, Shijiazhuang 050018, China; zhaoyongjie1982@126.com

[*]  Correspondence: jgwang123@hebust.edu.cn (J.W.); hfs_high@126.com (F.H.); Tel.: +86-0311-81668693 (J.W.)

**Abstract:** The piston rod of the hydraulic jack was a kind of artillery alloy structural steel of 40Cr and its surface was the chromium plating layer. The piston rod worked for a while; the defects of corrosion pit and peeling appeared on the chromium layer. A stereo-microscope, metallographic microscope, and scanning electron microscope (SEM) were used to observe the macromorphology, micromorphology, and microstructure of the failed parts. The composition analysis was performed by the energy dispersive spectroscopy (EDS), and a Vickers hardness tester was used to measure the hardness. The results showed that pores and penetrating cracks existed in the chromium layer, leading to the corrosion medium invading the interface and forming a corrosion source. Then, the corrosion intensified, resulting in the bubbling, cracking, and peeling of the chromium layer. More O and minor S elements were detected in the corrosion pit. Finally, the fracture of the chromium layer was a 45° angle destruction mode. The peeling of the chromium layer was caused by the pores and microcracks, working medium, and poor working environment. Some suggestions were put forward to prevent the peeling of the chromium layer.

**Keywords:** material failure and protection; piston rod; the chromium plating layer; peeling; corrosion





## 1. Introduction

As a standard hydraulic transmission device, the hydraulic jack is driven by the workforce or electricity. It is mainly used to lift or move heavy objects in shipyards, machinery factories, and coal mines. As an essential component in the hydraulic transmission system, the piston rod has a chromium plating layer on the surface. Still, it is exposed to corrosive environments such as the atmosphere, seawater, and mines for a long time and runs most frequently. After working for some time, the chromium plating layer would peel, leading to piston rod damage [1]. Piston rod damage is a failure mode with the highest maintenance cost, the longest maintenance time, and the highest maintenance value [2].

Chromium plating can effectively improve the surface hardness and wear resistance of shafts and rods, making it an essential surface treatment method [3]. However, some links will lead to the generation of residual tensile stress, such as hydrogen absorption, substrate surface treatment, and grinding, which will cause microcracks in the chromium plating layer [4]. Because of the electroplating process itself, the higher the hardness of the chromium plating layer, the greater the internal stress. In addition, the thicker the chromium layer is, the easier microcracks are produced. Yan [5] analyzed the reticular crack on the surface of the chromium plating layer of the 20CrNi3 piston rod and concluded that the reticular crack in the chromium plating layer is caused by the microcracks and high residual stress on the surface of the sample before chromium plating. Zhou [6] pointed out that when the internal stress of the chromium plating layer accumulates to a certain extent,

the chromium layer will be cracked. Moreover, the chromium plating layer needs grinding after chromium plating. If the grinding feed is too large, a significant residual tensile stress will exist in the chromium layer, resulting in microcracks. Huang [7] investigated the microcracks on the chromium plating layer of the shaft and rod surface and discovered that the microcracks are grinding cracks.

According to the current research and analysis, due to the manufacturing defects and damage in the use process, many penetrating microcracks exist in the chromium plating layer. Because of these penetrating microcracks, the corrosion resistance of the chromium plating layer will decline in the harsh and corrosive environment. Hu [8] analyzed the damage of the chromium plating layer on the surface of a naval gun barrel and concluded that the cracking and peeling of the chromium plating layer is the fundamental cause of barrel corrosion. In addition, pores on the surface of the chromium plating layer could also decline the corrosion resistance of the chromium plating layer. C. [9] analyzed the corrosion of a microporous chromium-plated surface in a concentrated water-electrolyte and concluded that the pores on the chromium-plated surface lead to different galvanic corrosion. The corrosive working medium would directly cause the failure of the chromium plating layer. K. [4] analyzed the premature failure of the rugged chromium-plated rotor of a downhole drilling motor. The results have shown that due to the corrosivity of drilling fluid, corrosion pits and falling-off appear on the surface of the chromium-plated rotor.

The piston rod studied in this paper was a part equipped with a hydraulic jack, and its surface was a milky white chromium plating layer. The piston rod worked for 2 to 3 years, and at the same time, the chromium plating layer appeared to have corrosion damage. The peeling defect of the chromium plating layer occurred on the piston rod, which would not only seriously affect the service life of the piston rod but also damage the sealing system and cause liquid leakage and failure to maintain pressure, eventually affecting the regular use of the hydraulic jack and causing a lot of losses. Therefore, failure peeling of the chromium plating layer of the piston rod was necessarily analyzed in order to reduce the loss. Aiming at the peeling phenomenon of the chromium plating layer on the piston rod, the macro- and micromorphology, microstructure, and properties of its failure parts were observed and detected. The reasons for peeling were discussed, and the improvement measures were put forward. This paper had a certain guidance effect on the chromium plating process and the piston rod protection.

## 2. Failure Analysis

The hydraulic jack used liquid as the working medium, transmitting power through the change of the sealing volume. The liquid used was the No. 4 resident solution, which was developed and manufactured by the Armory factory and was composed of ethylene glycol, distilled water, corrosion inhibitor, and other additives. The liquid was mainly used for artillery recoil devices, pressure balancing machines, and hydraulic jacks, which possessed excellent low-temperature and hydraulic transmission properties. The piston rod made a reciprocating linear motion in the hydraulic cylinder to lift heavy objects. The speed of the piston rod that contacted the sealing ring and guide sleeve was not fast during movement, as shown in Figure 1.

In this case, as shown in Figure 2a, the substrate material of the piston rod was 40Cr, which was a kind of artillery alloy structural steel, and its technical standard was YB476-64. Its length was 400 mm, and the diameter of its main part was 45 mm. The surface of the piston rod was a milky white chromium plating layer. The piston rod had different requirements for various parts during the chromium plating [10], and the thickness range of the chromium layer was 0.04–0.06 mm. The thickness of the chromium layer of its main part was 0.04 mm, and the thickness of the chromium layer in the head of the piston rod was increased to 0.06 mm. The surface finish of the central part of the piston rod was ▽7, and the surface roughness was calculated as Ra6.3–Ra12.5 μm.

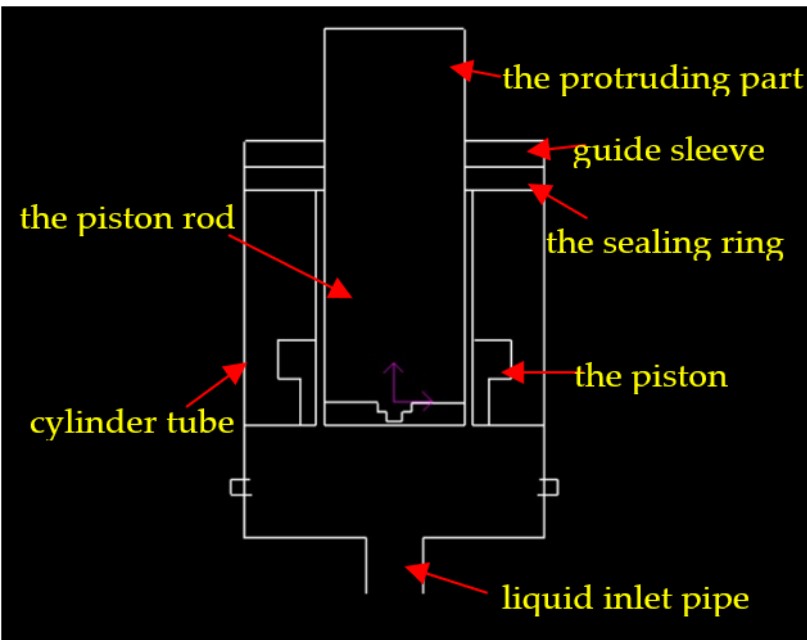

**Figure 1.** A schematic diagram of the motion of the piston rod.

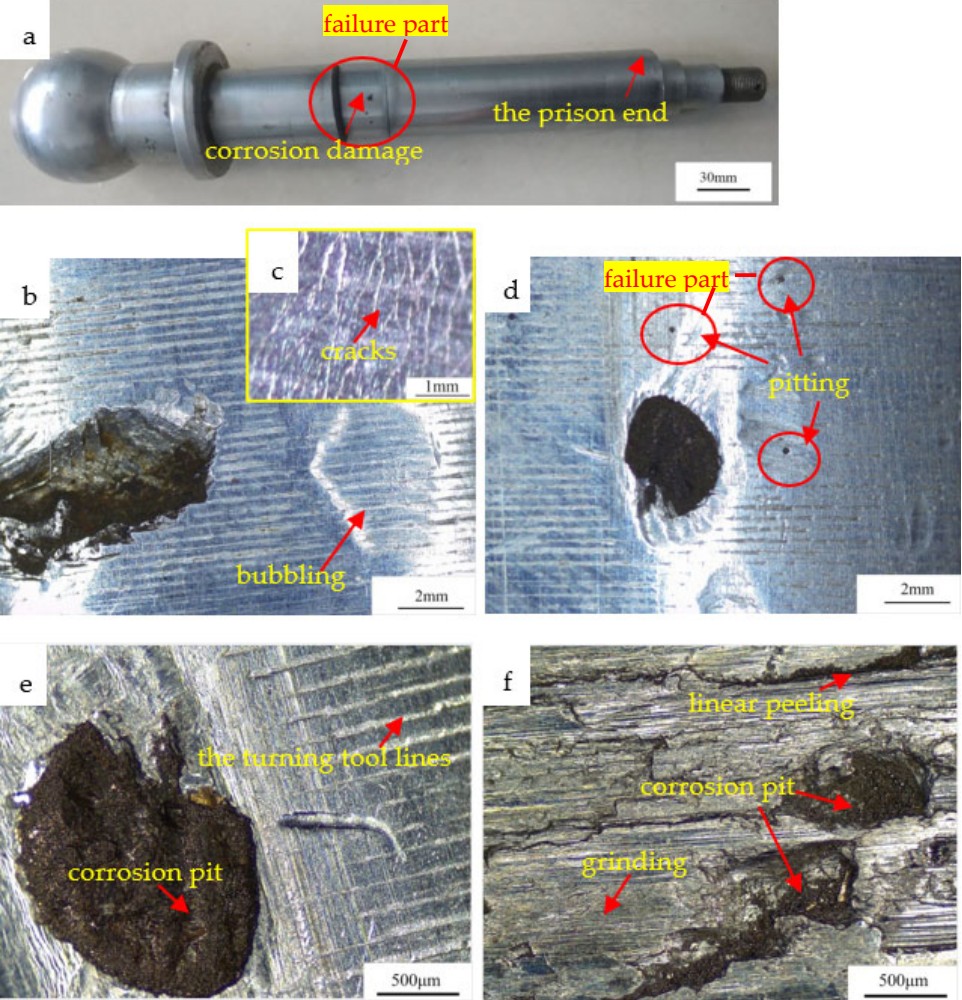

**Figure 2.** Macroscopic morphology observation: (**a**) piston rod; (**b**) bubbling of the chromium plating layer; (**c**) magnified surface of bubbling; (**d**) pitting; (**e**) corrosion pit; (**f**) grinding and peeling.

The hydraulic jack was used in the military factory and was an accessory piece for the artillery. After the piston rod worked for a while, the corrosion pit and peeling occurred on the chromium plating layer in the middle area of the piston rod. No corrosion damage was found on the chromium plating layer close to the piston end, as shown in Figure 2a. Figure 2b shows the bubbling of the chromium plating layer in which a number of cracks were distributed on its surface in Figure 2c. Some pitting was distributed in the middle area of the piston rod in Figure 2d. In addition, there were also two defects in the middle area: corrosion pit and linear peeling, as shown in Figure 2e,f. Through measurement and observation, the areas of the corrosion pits were different, and the pit was black. The chromium plating layer completely fell off and the substrate was exposed. It could also be seen that the chromium plating layer on the edge of the corrosion pit was rough and uneven, and the clear turning tool lines were left by a profiling deposition. After grinding, the thickness of the chromium plating layer was reduced. In addition, the shape and density of the cracks in the chromium plating layer were changed, and linear peeling along the cracks occurred. The linear peeling was about 3 mm in length and 0.1 mm in width, distributed along the circumferential direction in which black substances were also distributed.

### 3. Materials and Methods

The failure part of the piston rod was cut by WEDM technology to obtain the surface and section area of an appropriate size. The chemical composition of the substrate was measured by an optical emission spectrometer. The macroscopic morphology of the damaged site was observed by the Lei-ca stereomicroscope. The specimens were ground by sandpaper with 400–1200 eyes, polished by 1 μm polish paste, corroded by 4% nitrate alcohol, and the substrate structure was observed using a ZEISS inverted metallographic microscope (OM). The Vickers hardness tester of TMVS-1 was used to test the hardness of the chromium plating layer and substrate, which was applied at 200 g load for 10 s and measured three times, and the average value was taken. The surface morphology of the failure part was observed by a scanning electron microscope of TESCEN-VEGA3, and the composition was analyzed by energy dispersion spectroscopy (EDS).

### 4. Results

The chemical composition of the substrate was determined, as shown in Table 1. It could be seen that all kinds of elements met the standard requirements.

**Table 1.** Chemical composition of 40Cr (wt%).

| Element | C | Cr | Si | Fe | Ni | Cu | Mn | P | S |
|---|---|---|---|---|---|---|---|---|---|
| Wt% | 0.41 | 0.97 | 0.29 | Bal. | 0.047 | 0.049 | 0.64 | 0.011 | 0.01 |

As shown in Figure 3, the bonding state between the chromium plating layer and the substrate was observed through a metallographic microscope. According to Figure 3a, the thickness of the chromium plating layer in the middle of the piston rod was 43.15 μm, which was physically combined with the substrate material. In Figure 3b, cracks penetrating the chromium plating layer were found, which could provide a channel for the corrosive medium. In Figure 3c, the substrate had been corroded, and corrosion sources existed at the interface between the chromium plating layer and the substrate. At this time, the chromium plating layer was not peeled off. The corrosion in Figure 3d extended further and the chromium plating layer cracked and peeled off, making the corrosion medium easier to penetrate, and the corrosion intensified. In Figure 3e, part of the chromium plating layer had separated from the substrate, resulting in the steel substrate being completely exposed. The chromium plating layer belonged to the cathodic coating. The combination of a large cathode and a small anode aggravated the corrosion, resulting in the corrosion area of the substrate being larger. In Figure 3f, the steel substrate was further corroded, resulting in the fracture and peeling of the chromium plating layer at the edge of the corrosion pit.

The corrosion of the steel substrate proceeded along the width and depth direction of the corrosion pit. It could be seen that the pit depth was 137 μm in Figure 3f.

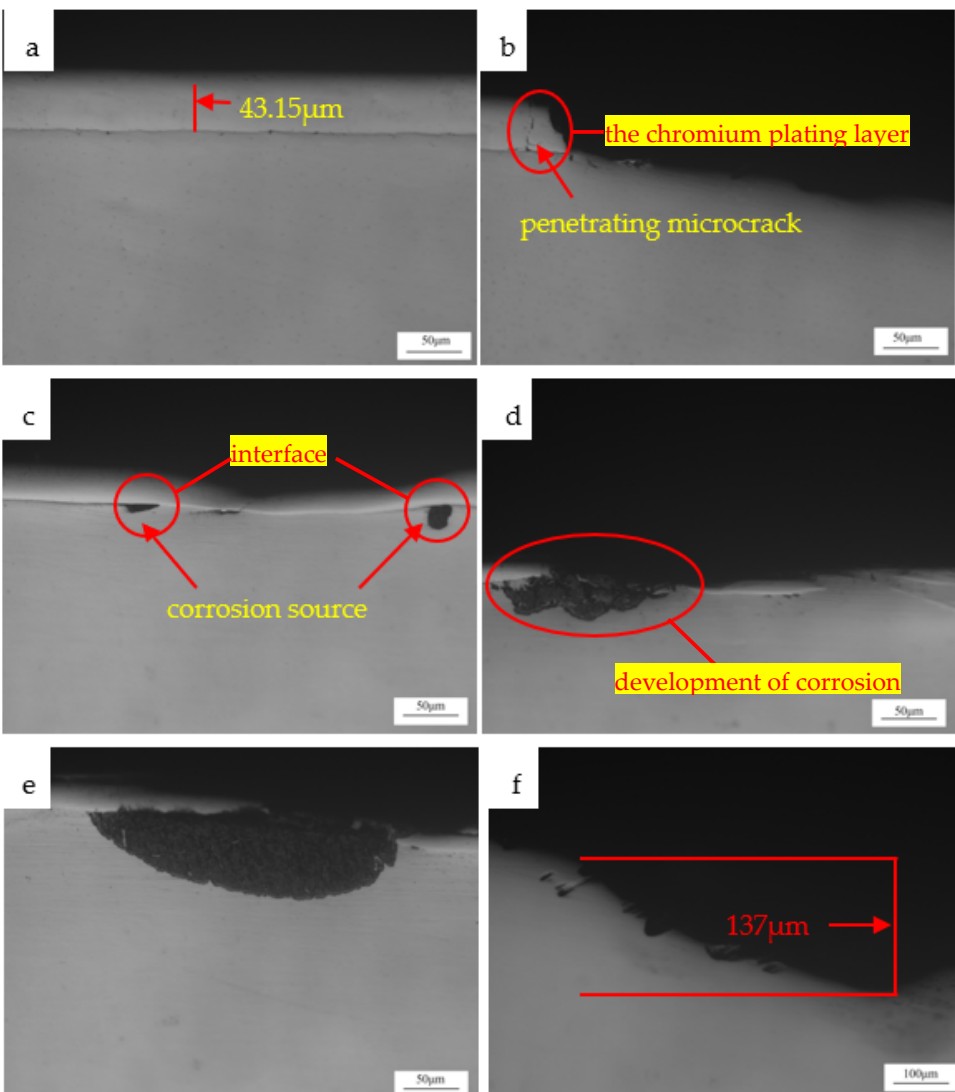

**Figure 3.** OM observation on the bonding state of the chromium plating layer and substrate: (**a**) intact part; (**b**) penetrating microcrack; (**c**) corrosion source; (**d**) development of corrosion; (**e**) aggravation of corrosion; (**f**) the cross-section of corrosion pit.

The cross-section of the test piece was ground and polished so that the structure could be observed after the corrosion. It could be seen from Figure 4a that the substrate structure was tempered sorbite and that fine, uniform, and white cementite were distributed in the ferrite substrate. Here, the carbides were spherical and stable. The hardness of the substrate was 240HV0.2, and the chromium plating layer was 600HV0.2 by using a Vickers hardness tester, which met the standard requirements. In Figure 4b, a crack network existed in the chromium plating layer, and some cracks penetrated the chromium plating layer. Figure 4c shows the cross-sectional structure morphology of the corrosion pit where no cracks occurred near the pit, and the structure was typical.

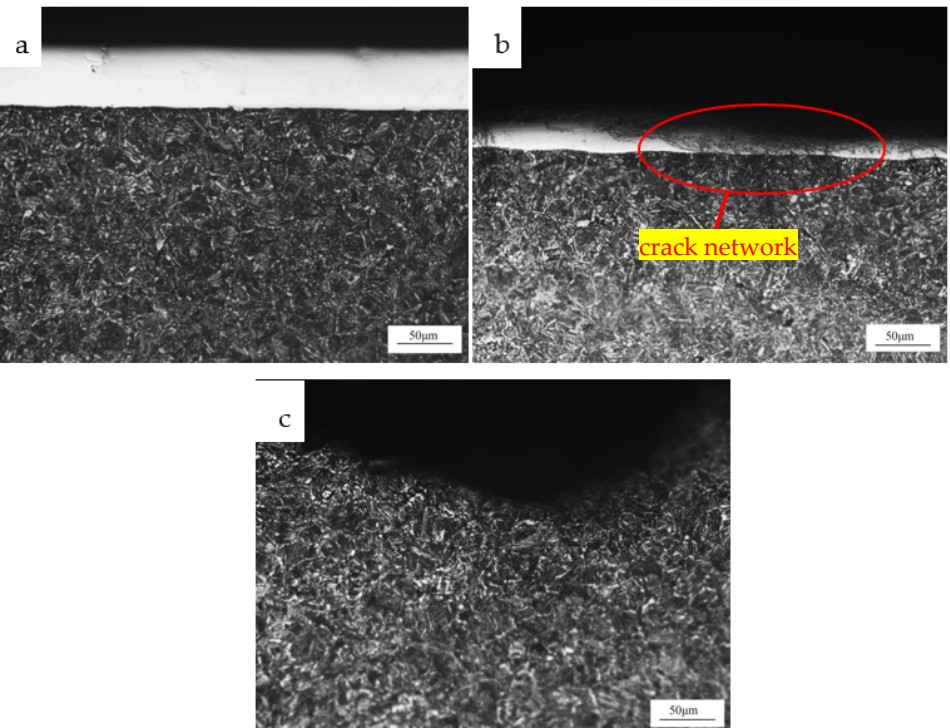

**Figure 4.** OM observation of structure: (**a**) substrate; (**b**) crack part; (**c**) the cross-section of corrosion pit.

Figure 5 is an SEM observation of the surface morphology. The turning tool lines and micropores were distributed on the surface of the chromium plating layer in Figure 5a. It could be seen from Figure 5b that cracking was distributed on the surface of the chromium plating layer. The chromium plating layer in Figure 5c was seriously cracked and partially peeled, and corrosion pits also occurred. The chromium plating layer was broken and peeled in a large area, and corrosion products could be seen in Figure 5d. Point A was a spherical corrosion product and the composition of point A was detected. In Table 2, it could be seen that Fe was the main element, and the content of the O and S was, respectively, 11.49 and 0.52%, indicating iron oxides and sulfides were formed here. The composition analysis of point B in Figure 5d showed that Cr was the main element, and the content was 85.73%, suggesting the chromium plating layer. In addition, Point B also contained the elements C and O. Element C was caused by the residual dry lubricating oil on the surface of the piston rod, and element O should be the result of the oxidation of the chromium plating layer in the air. Figure 5e shows the morphology of a corrosion pit where the surface was rough and uneven, with bulges and cracks. Point C was a bulge and the composition analysis of point C showed that Fe was the main element, and the content of O and S was, respectively, 11.18 and 0.6%, indicating that the substrate was seriously corroded here. Figure 5f showed a corrosion extension where large cracks, substrate peeling, and pits were distributed. Point D was a spherical bulge that was distributed more on the substrate. The composition analysis showed more O and S elements and was parallel to point C, indicating the substrate was seriously corroded. Point E was a pit, and EDS showed that the content of the O element was 26.20%, indicating that the corrosion here was the most serious.

Figure 6 shows an observation of the SEM morphology of the cross-section. Penetrating cracks were found in Figure 6a, which would provide channels for the corrosive medium. A fractured chromium layer was also found and the cleavage surface was about 45° from the axis of the rod. It could be seen that the corrosion products were distributed in the pit, and corrosion expansion extending to the width and depth direction of the corrosion pit was identified in Figure 6b.

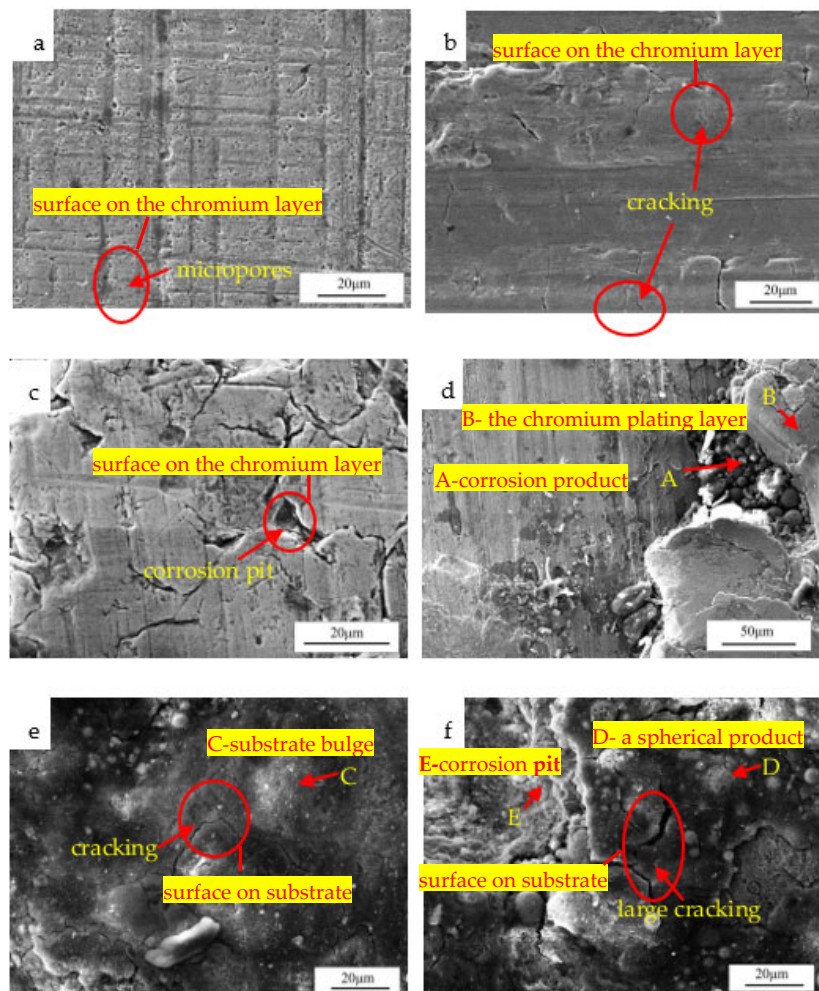

**Figure 5.** SEM micromorphology observation: (**a**) chromium plating layer; (**b**) cracking; (**c**) corrosion pit; (**d**) peeling of the chromium plating layer; (**e**) exposed substrate; (**f**) corrosion extension.

**Table 2.** Chemical composition of A–E in Figure 5 (wt%).

| Point | C | Cr | Fe | O | Si | S | Mn |
|-------|------|-------|-------|-------|------|------|------|
| A | 10.49 | 1.88 | 75.29 | 11.49 | 0.23 | 0.52 | 0.10 |
| B | 6.73 | 85.73 | 0 | 7.54 | 0 | 0 | 0 |
| C | 8.31 | 0.45 | 77.49 | 11.18 | 0.37 | 0.60 | 0.13 |
| D | 9.82 | 0.27 | 78.43 | 10.39 | 0.32 | 0.45 | 0.32 |
| E | 8.69 | 0.80 | 61.99 | 26.20 | 0.41 | 0.18 | 0.29 |

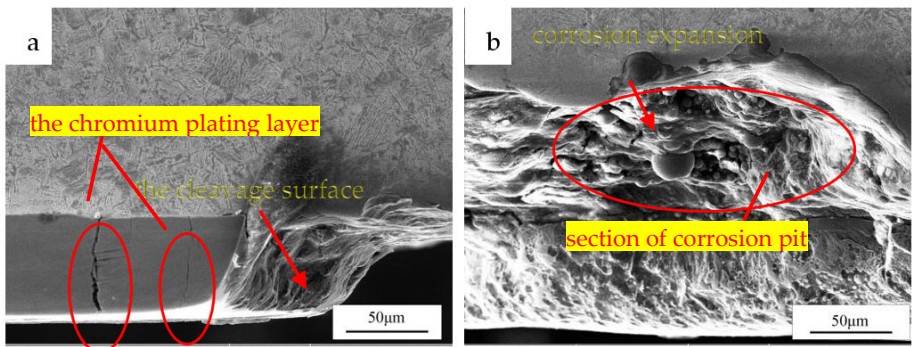

**Figure 6.** SEM morphology observation: (**a**) a fractured chromium layer; (**b**) section of corrosion pit.

## 5. Discussion

The substrate was corroded in a corrosive medium because of penetrating cracks and pores in the chromium plating layer. As the corrosion intensified, the chromium plating layer would bubble, crack, and peel off [11].

For the network cracks and penetrating cracks in the chromium plating layer, it should be that there was a considerable internal stress in the chromium plating layer. In addition, the chromium plating layer had high hardness and brittleness, which made it easy to produce microcracks [12]. Generally, after chromium plating, there was a specific residual tensile stress in the chromium plating layer. The reason for this was that the layer's equilibrium spacing between atoms was smaller than the substrate, resulting in additional tensile stress on the chromium plating layer [5]. In addition, chromium was deposited in the electroplating process, accompanying the generation of a large amount of hydrogen, which would penetrate the chromium layer and react with the chromium to form CrH, which was unstable and brittle. Some studies have shown that the internal stress on the surface of the chromium layer and the stress caused by hydrogen diffusion to the chromium layer are surface residual tensile stresses. Cracks would appear when the stress exceeded the fracture strength of the chromium layer material [13]. At the same time, to achieve the required contour accuracy, the chromium plating layer also needed to be machined after electroplating, which was usually cylindrical-forming grinding. The superposition of the grinding force and internal stress in the chromium layer would change the density and shape of the cracks, aggravating the development of cracks in depth and width and leading to penetrating cracks in the chromium plating layer, as shown in Figures 2b and 5a. The turning tool lines on the surface of the piston rod were seen in Figure 1e, and the residual stress of the turning tool lines would also affect the cracking of the chromium plating layer [8]. In addition, loose pores on the surface of the chromium plating layer could also cause corrosion, as shown in Figure 2d. Pores would be created in the chromium plating process, which were generally small in size and high in distribution density [14], as shown in Figure 5a.

The chromium plating layer was physically combined with the substrate material in which the binding force was weak. In addition, the surface of the chromium plating layer had loose pores and many microcracks. The corrosion mechanism was the same whether the corrosion occurred through microcracks or pores. The piston rod made a reciprocating linear motion in the hydraulic cylinder. When the piston rod was in a protrudent state, the protruding part of the piston rod was inevitably affected by corrosive media and dust pollutants in the bad environment. Corrosion damage occurred. However, the piston end was in a hydraulic cylinder because of the maximum protrudent requirements and was not exposed to air and without corrosion. The working medium was the No. 4 resident solution, and its main ingredient was water. However, the water was easy to evaporate, resulting in the rupture of the water film. The hydraulic jack was an accessory piece to the artillery. Due to the artillery gas, the oxygen and the sulfur atmosphere were mixed into the water film and became a corrosion medium. The pores and penetrating cracks of the chromium plating layer provided channels for the corrosive media, forming corrosion sources [15,16]. The substrate beneath the chromium plating layer could contact the corrosion solution and would be corroded. With the development of corrosion, corrosion products gradually increased and took on a spherical shape, as shown in Figure 5d. Under the action of the O and S elements, FeS and $Fe(OH)_2$ could be generated. The specific volume of these corrosion products was much larger than that of the iron, so expansion was generated. When the expansion force was greater than the binding force between the chromium plating layer and the substrate, the chromium plating layer would bulge and gradually increase to form bubbling, as shown in Figure 2b. As a result, cracking would occur, making it easier for the corrosive medium to penetrate, as shown in Figures 2c and 5c. As the corrosion intensified, the bubbling cap would peel off, and the corrosion pit full of corrosion products was exposed, as shown in Figures 2e and 5e. After creating the corrosion pit, the iron became the anode, and the chromium plating layer was the cathode. The cathode area was much larger than that of the anode. Combining a large cathode and a small anode made

the substrate corrode rapidly. The substrate would be further corroded, resulting in the fracture and peeling of the chromium plating layer at the edge of the corrosion pit. The fracture mode of the chromium layer was a 45° angle destruction, as shown in Figure 6a, and was equivalent to solid support bending destruction. One end of the rod was fixed, and the displacement was completely constrained in each direction. The cleavage part of the chromium plating layer was equivalent to the solid support end of the rod, the corrosion product under the cracked chromium layer expanded due to the volume, and the bending load was applied to the cracked chromium layer. Finally, the corrosion continued to expand to the depth and width direction of the corrosion pit, as shown in Figures 3f and 6b.

Linear peeling distributed along the circumferential direction occurred due to cylindrical-forming grinding and fine machining. The grinding feed was large enough to produce large residual tensile stress. The superposition of residual tensile stress and internal stress intensified the crack development of the chromium plating layer. During fine machining, the turning tool lines were in the circumferential direction and produced internal stress. Therefore, when the stress superposition exceeded the fracture strength of the chromium layer, cracking occurred along the direction of the turning tool lines and was linear. Under the action of corrosion, linear cracking expanded, the corrosion products were exposed, and linear peeling appeared.

The following suggestions were put forward for the peeling of the chromium plating layer on the piston rod:

1.  Reduction in the brittleness of the chromium plating layer. The brittleness of the chromium plating layer was due to the plating process itself. The chromium plating layer with no cracks and good toughness could be obtained by appropriately increasing the electroplating temperature and reducing the current density [17]. Dehydrogenation treatment and minor grinding feed could reduce the internal stress in the chromium layer and improve the quality of the coating.
2.  Techniques for increasing the corrosion resistance of the chromium plating layer. The electroplating process of the piston rod caused a large number of micropores and microcracks in the chromium plating layer. The substrate surface required pretreatment and flaw detection before plating so as to reduce the defects in the chromium plating layer. The pore-sealing technology after plating or employing nickel-chromium composite plating could greatly improve the corrosion resistance of the chromium plating layer [6,18].
3.  Safeguarding during using process. The working media needed to be converted to hydraulic oil. Although the No. 4 resident solution had a certain hydraulic transmission performance, its main ingredient was water, which caused rust corrosion.
4.  Advanced protection method for the piston rod. Laser cladding repaired the damaged piston rod in a local cladding manner, omitting the most expensive plating process, and had the advantage of green environmental protection. In addition, laser cladding had the characteristics of a metallurgical combination, dense coating, and was better suited for the protection of the piston rod of a hydraulic jack [19].

## 6. Conclusions

Through the failure analysis of the chromium plating layer on the surface of the piston rod of the hydraulic jack, the following conclusions were drawn:

The chromium plating layer was physically combined with the substrate, and the substrate structure was tempered sorbite. The hardness of the chromium plating layer was 600HV0.2, and the hardness of the substrate was 240HV0.2. Penetrating cracks and pores occurred in the chromium plating layer and would provide channels for the corrosive medium. The EDS detection showed that the severely corroded parts contained more O and S elements.

Corrosion sources were formed at the interface between the chromium plating layer and the substrate because of the working medium, artillery gas, penetrating cracks, and pores. The chromium layer bubbled, cracked, and peeled due to the intensified corrosion.

Corrosion pits were subsequently formed and extended to the depth and width direction of the substrate, and the fracture of the chromium layer was a 45° angle destruction mode.

Microcracks and pores existed inevitably in the chromium plating layer. To improve the corrosion resistance of the chromium plating layer, the brittleness of the chromium plating layer needed to be reduced, and the substrate surface required pretreatment and flaw detection before plating. The pore-sealing technology after plating or nickel-chromium composite plating could be used to improve the quality of the coating. In addition, the working medium required adjustment. The laser cladding was an advanced method for the protection of the piston rod of the hydraulic jack.

**Author Contributions:** Data curation, Q.Z., W.S. and Y.Z.; Formal analysis, Q.Z. and W.S.; Funding acquisition, J.W.; Investigation, J.W.; Writing—original draft, Q.Z.; Writing—review & editing, J.W., W.S., F.H. and Y.Z. All authors have read and agreed to the published version of the manuscript.

**Funding:** This work was supported by the Military civilian science and Technology Collaborative Innova-tion Project (Grant No. 20351801D), S&T Program of Hebei (Grant Nos. 20351801D and 20564401D) and Key projects of Hebei Provincial Department of Education (Grant No. ZD2020189).

**Institutional Review Board Statement:** Not applicable.

**Informed Consent Statement:** Not applicable.

**Data Availability Statement:** The raw/processed data required to reproduce these findings cannot be shared at this time as the data also form part of an ongoing study.

**Conflicts of Interest:** We declare that we do not have any commercial or associative interest that represents a conflict of interest in connection with the work submitted.

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
