# Peer review of "Failure Analysis of Chromium Plating Layer on the Surface of the Piston Rod of the Hydraulic Jack"

_coatings, doi:10.3390/coatings12060774_

Round 1
Reviewer 1 Report
The manuscript reports the corrosion resistance of chromium-plated layers on piston rod, but it lacks any new findings or improvements.
(line 89, page 2) "plating layer with a thickness of 0.04-0.06mm"
The variation in the plating thickness is too large to be considered comparable. The authors fail to eliminate controllable uncertainties, which prevents a fair comparison between effects from difference sources, e.g. defects, residual stress, binding, etc.
(lines 327-339, page 8) "The following ..."
This paragraph summarizes what is found in this manuscript: out of 4 points mentioned, two are from literature (1 and 3). The other two are nothing but speculation without definitive and quantitative results supporting the arguments.
All in all, I do not see anything new in the manuscript.
Author Response
We would like to thank you for your careful reading, helpful comments, and constructive suggestions, which have significantly improved the presentation of our manuscript. For responses to comments, please see the attachment.

Reviewer 2 Report
- No mention in the abstract about the type of Position Rod material.
- In the (Failure Analysis) on what basis were the components of the corrosion medium liquid determined.
- Is the thickness of the (Chromium Plating) measured within this range (0.04-0.06 mm) or according to a reference ?.
- Why no corrosion occurred at the position end, needs to be reinforced with a discussion.
- Explanation of why Linear Peeling appears distributed along the circumferential direction .
- The test time (work) and the corrosion period were not indicated.
- From Table No. (2), why this different in the values ​​of Point B compared to the rest, it is required to clarify this behavior.
- In the conclusions, it is preferable to refer to the method of coating and adjusting the coating parameters that have an effect on the quality of the coating and its resistance to corrosion conditions.
Author Response

(The authors gave the same response as above.)

Reviewer 3 Report
The article under review is devoted to identifying the causes of the destruction of the chromium coating of the hydraulic piston rod. The article has a practical focus, such studies are often useful to manufacturers and consumers of products, including in resolving disputes and establishing the causes of equipment failure. During the review, I had a few questions and comments:
1) The quality of the photos in Figure 1 needs to be improved. Now even the ruler in the photo is not clearly visible.
2) In the «Discussion» and «Conclusions» it is said about a large number of micropores in the coating. However, the photographs (Fig. 2) do not show micropores. I recommend that the authors of the paper provide a photo of the coating with micropores.
3) Did I understand correctly that the main recommendation for improving the chrome coating is to replace it with a coating obtained by laser cladding? Why do the authors of the article think that there will be no such problems with laser cladding? In laser cladding, coating cracks are also a common defect.
4) As I understood from the article, the cause of the violation of the continuity of the coating and subsequent corrosion is caused by the formation of cracks due to the fragility of the chromium coating. What recommendations can the authors of the article give to reduce the fragility of the chromium coating?
Author Response

(The authors gave the same response as above.)
